# A Study on Fatigue Crack Propagation for Friction Stir Welded Plate of 7N01 Al-Zn-Mg Alloy by EBSD

**DOI:** 10.3390/ma13020330

**Published:** 2020-01-10

**Authors:** Wenyu Liu, Dongting Wu, Shuwei Duan, Tao Wang, Yong Zou

**Affiliations:** 1School of Materials Science and Engineering, Shandong Engineering & Technology Research Center for Modern Welding, Shandong University, Jinan, Shandong 250014, China; liuwenyu01@foxmail.com (W.L.); wudongting@sdu.edu.cn (D.W.); shuwei01@163.com (S.D.); 2Key Laboratory of Liquid-Solid Structural Evolution and Processing of Materials, Ministry of Education, Shandong University, Jinan, Shandong 250014, China

**Keywords:** Al-Zn-Mg alloy, fatigue crack propagation, friction stir welding, EBSD

## Abstract

EBSD (electron backscattered diffraction) was used to study the fatigue crack propagation mechanism in a friction stir welding joint of a 15 mm-thick 7N01 aluminum alloy plate. Crack tips with detailed features were clearly characterized by EBSD images. The plastic zone caused by crack was small in the stir zone. Due to the fine grain strengthening in the stir zone, there were several lattice distortion regions which were observed in the BC (band contrast) map but disappeared in the SEI (secondary electron image). In the stir zone, fatigue crack tends to awake and grow along grain boundaries, and propagate with little deformation of the grains. When the crack tries to grow across a boundary, the deformation of the plastic zone at the crack tip shows little correlation to the cyclic loading direction. However, the plastic zone in base metal, the rolled plate, is large and continuous, and no obvious lattice distortion region was found. According to Schmidt factor, the base metal near crack is fully deformed, lots of low angle boundaries parallel to the cyclic force can be observed. The base metal showed a better ability for fatigue crack propagation resistance.

## 1. Introduction

Since FSW (friction stir welding) was invented by TWI (The British Welding Institute) in 1991 as a solid phase joining technology, it has been a method for the welding of nonferrous metals and alloys. As a thermo-mechanical coupling process, FSW connects materials on atom level by plastic deformation [1]. For those metals with low melting temperatures, such as aluminum and magnesium, FSW joints usually show high performance without defects created during fusion welding, and it has been widely used in the area of aerospace and transportation [2,3].

A majority of failures of engineering materials and structures are a result of fatigue. Although our understanding of fatigue of materials and structures has grown significantly over the years, failure from metal fatigue is still an ongoing engineering problem. The fatigue performance of FSW joints and the mechanism of fatigue have become more important with the wide usage of FSW [4]. AA 7N01 is a novel Al-Zn-Mg series aluminum alloy, mainly used for weight reduction components on high strength structures. In general, the evaluation and prediction of fatigue properties of materials under service are based on large quantities of experiments, and mostly by experience. However, the mechanism of fatigue is not fully understood. The fatigue of materials after friction stir welding still needs more research because of the varied microstructure of the joint.

At present, there are many kinds of research methods for studying fatigue cracks. Many studies have reported a mechanism through fracture morphology combined with undamaged microstructural features [5,6,7,8,9,10]. However, this method observes crack indirectly without the behavior of material near the crack, and the interpretation of fracture morphology often requires experienced researchers. On the other hand, with the development of EBSD (electron backscattered diffraction) and the SEM (scanning electron microscope), more and more researchers focus on the crack behavior and the deformation around it, especially using EBSD recently [11,12,13,14,15,16,17]. Gupta, V.K. et al. [13] researched the role of particle clusters during fatigue crack initiation stage by EBSD of fracture morphology. This method requires a FIB (focused ion beam) to make a flat area. And in their another paper, the mechanism of fatigue crack initiation was studied through a more economical way, mechanical polishing. Li, L.L. et al. [15] studied the behavior of grain boundaries and slip bands under fatigue of Cu bicrystals. As is shown, EBSD shows great potential for us to further understand the mechanism of fatigue crack initiation and propagation. However, previous studies mostly concentrate on the fatigue crack initiation stage with low ∆K (stress intensity factor) level, and the EBSD acquisition is limited to a small region.

Fatigue failure consists of three stages: initiation, propagation, and fracture, as is shown in Figure 1. This paper shows detailed EBSD results with relatively large region for fatigue crack propagation stage under higher ∆K. Material’s behavior near crack was observed, in order to study the mechanism of fatigue crack propagation.

## 2. Materials and Methods 

A 15-mm-thick 7N01 aluminum alloy butt welded plate was used for this study. The welding parameters were 400 r/s for rotation speed and 200 mm/min for moving speed. The chemical composition tested with electric spark-spectrographic is shown in Table 1. Tensile tests were performed using the Zwick Roell Z250 universal testing machine with a laser extensometer (Ennepetal, Germany). The main parameter of tensile specimen is illustrated in Figure 2. Metallography and EBSD samples were prepared using electrolytic polishing. For metallography observation, samples were etched with Keller’s reagent, whose composition is as follows: 1 mL HF, 1.5 mL HCl, 2.5 mL HNO_3_, and 95 mL H_2_O. Metallography is observed by VHX-500F KEYENNCE digital microscope (Osaka, Japan). EBSD samples were polished with a MS-3010D DC power supply and an electromagnetic stirrer, with a solution consisting of ethanol (C_2_H_5_OH): perchloric acid (HClO_4_) = 9:1. The optimized parameters for electrolytic polishing were 18 V, approximately 0.4 A current when it is stable, an 5 × 25 mm polish surface, and 30–45 s at a temperature of 5–10 °C. A relatively low voltage can save the detailed features from damage. However, the parameters can vary slightly because of the different crack opening conditions, and it may take several specimens to polish repeatedly to acquire EBSD images with high quality and large region. We used a JSM-7800F scanning electron microscope (SEM) (Tokyo, Japan) and Oxford NordlysMax3 system to acquire EBSD data, and then processed them with Channel 5 commercial software (v5.12.67.0) without noise reduction in order to maintain the authenticity of cracks.

The FCP (fatigue crack propagation) test was performed according to ASTM E647 with a CT (compact tension) specimen using GPS-100 high-frequency fatigue testing machine at room temperature (25 °C) in air. Specimens were designed with 50 mm W and 15 mm thickness. The notch was made by electrical discharge machining. FCP tests were conducted with increasing K under constant load with a parameter of 0.7 stress ratio R, about 107 Hz frequency. P is the force during cyclic load. The force P corresponds to the stress intensity factor K through the equation:(1)$$K=\frac{P}{B\sqrt{W}}\frac{\left(2+\alpha \right)}{{\left(1-\alpha \right)}^{3/2}}\left(0.886+4.46\alpha -13.32\alpha^{2}+14.72\alpha^{3}-5.6\alpha^{4}\right)$$
where α = a/W. ΔK corresponds to ΔP, which is the range of cyclic force. Based on Paris formula, the FCP rate can be implied as da/dN = C(∆K)^m^, where ∆K is stress intensity factor; C and m are two constants determined by materials properties. The ΔK plays a role of connecting different test conditions. Considering the lower yield strength of the joint according to tensile results and time efficiency, set P_max_ 10.7 kN for base metal and 8.7 kN for stir zone. The initial K_max_ was 510 MPa$\sqrt{m}$ for base metal and 415 MPa$\sqrt{m}$ for the stir zone. According to experience, both the base metal and stir zone are in propagation stage with their unique C and m. No process for the deep strings on the top surface marked inevitably by FSW to make the test condition more realistic. Considering the contingency in the test, an FCP test in each region was performed three times with three specimens.

For all EBSD tests, the sample coordinate system is shown in Figure 2, which illustrates each specimen′s position schematically as well. EBSD specimens for crack tip were cut from the center of the thickness where is usually considered to be the longest crack propagation spot and the most restrained plane; meanwhile, the crack here propagates most likely in plane strain state. In order to avoid the effects by wide opened crack, tests were performed at the spot when crack propagated to 5 mm, where the ∆K is 662 MPa$\sqrt{m}$ for base metal and 538 MPa$\sqrt{m}$ for stir zone.

## 3. Results and Discussion

### 3.1. Metallography

Figure 3 is the macrostructure of an FSW joint divided into different regions, including SZ (stir zone), wherein the material is stirred fiercely; TMAZ (thermo-mechanical affected zone): the material in this region is affected by both heat and deformation; HAZ (heat affected zone), the region where the material influenced by heat only and shows no specific boundary with base metal. The tool rotates constantly while moving forward; the side where the tool spins to a direction consistent with its moving direction is named AS (advancing side), and the other side is RS (retreating side).

### 3.2. The Microstructural Features of Stir Zone

Figure 4 illustrates the microstructures of the regions corresponding to the black rectangles in Figure 3 and the detailed center area of the SZ with a recrystallized fraction. Each region shows an area of 600 × 400 µm for 0.5 µm steps. Although a thicker plate, of 15 mm, was studied, the microstructure variation showed a similar trend to that in studies by Ahmed, M.M.Z. et al. [18] and Tao, W. et al. [19]. The center area consists of fine equiaxed grain with an average diameter of 6.3 µm; small angle boundaries distribute uniformly. It can be seen from the recrystallization component diagram that most of the recrystallized grains are blue with no deformation features, which indicates that the center region was fully dynamically recrystallized.

### 3.3. Tensile Properties.

The results of tensile are shown in Figure 5 and Table 2. The joint has a lower yield strength (σ_y_) and a much higher elongation (δ). The ultimate tensile strength (σ_UTS_) of the joint is very high, and reaches 97% of the base metal. That is probably caused by the fine equiaxed grain and the natural aging behavior. The fracture of tensile specimen is from stir zone to the HAZ of retreating side. The results show a trend of decreasing strength and increasing elongation, which is similar with the study of Zhao, Y. X. et al [20]. Their σ_y_, σ_UTS_, and δ for base metal are 325.67 MPa, 463.54 MPa and 15% respectively. While their σ_y_, σ_UTS_, and δ for the joint are 253 MPa, 402 MPa, and 16.3% respectively.

### 3.4. The Results of FCP Test

The results of FCP are plotted in Figure 6, showing valid a–N (crack length-number of cycles) data according to ASTM-E647. We processed the a–N data with the secant method to obtain the lg(da/dN)-lg∆K scatter plot, and then obtained a straight line through linear fitting; the results are shown in Figure 6. Therefore, the Paris formulas are da/dN = 8.51 × 10^−10^ ∆K^2.19^ and da/dN = 6.76 × 10^−13^ ∆K^3.34^ for stir zone and base metal respectively.

Obviously, FCP rate in base metal is slower than that in stir zone; thus, there is a stronger resistance in the base metal, which is consistent with many previous reports [10,21,22,23]. There are lots of fine equiaxed grains in the stir zone. Normally, due to the fully dynamic recrystallization of the stir zone, the large grain boundary density in stir zone is much higher than that in base metal, which can be initiation spots for fatigue crack [8,15,24,25].

### 3.5. EBSD Features for the Crack Tip on the Base Metal

Maps in Figure 7 and Figure 8 are different methods to analyze the crack tip in base metal, showing a region of 1218 × 837 µm at 1.5 µm step. Some second phase particles lying on base metal can be observed in Figure 7. Also, the contour of the crack in SEI (secondary electron image) can match with that in BC (band contrast) well, and no obvious lattice distortion zone is shown. Figure 9 shows the EDS (energy dispersive spectrometer) characteristics for several types of second phase particles with typical shapes in the 7N01 base metal. It is clear that the particles with a shape of cylinder protrusions are Si-rich particles and those ones with pit shaped are Fe-rich particles. It can be seen some particles signed with yellow circles are Si-rich particles and one of them has affected the crack propagation. The crack propagation enters the crystal from the grain boundary, and the propagation path shifts. Meanwhile, the Fe-rich particles circled with white show little influence on crack propagation path. Generally, second phase particles are defects during fatigue, since some weak heterogeneous interfaces can become crack initiation spots easily. Although some research based on simulations proposes a strengthening effect of particles [8,26,27,28], more evidence is needed to show how second phase particles work during fatigue. In addition, the tip of the crack also shows the branching: one zigzag grows into the next grain; the other tends to continue along a grain boundary. And there is a crack awakening spot in front of the crack; it might expand backwards and merge with former cracks.

In the BC map of Figure 7, two large grains can be observed around the crack, which are brighter than other grains because of the higher quality of diffraction pattern, a sign of little deformation and inner stress. These larger grains are usually formed by recrystallization and become harder, stronger, and more brittle, while most other grains are fully deformed with lots of low angle boundaries and a better property of toughness. In the large grain, slip bands (yellow arrows), caused by a relatively high load and the strong resistance for deformation, are shown clearly, and the crack grows along one of them. When the crack propagates with high resistance, such as trying to cross a grain boundary, it will create a plastic zone. The weakest points in the plastic zone, grain boundaries for example, can probably become sources of cracks. So it is possible for the crack to initiate at a grain boundary (white arrow) and propagate in the opposite direction, considering the intersection of a crack and a slip band (white circle).

Figure 8 shows the IPF-X0, IPF-Y0, IPF-Z0, KAM (local misorientation), and Schmidt factor distribution. The Schmidt factor is a ratio of the resolved shear stress with the main stress. To obtain Schmidt factor map, set the force parallel to fatigue load with the slip system {111} <101>. Deformation can become harder as its degree increases, and the Schmidt factor decreases during the process. We can see a similar pattern of arrangements for low angle boundaries—parallel lines in almost every grain while at different angles, some of which are illustrated schematically. A low Schmidt factor suggests fully deformed grains, such as those green ones with lots of low angle boundaries parallel to load direction, but those red ones retain high Schmidt factor values, such as the two large recrystallized grains hardly deformed, in spite of cracked slip bands, due to their harder, stronger, and more brittle nature. Also, the KAM map shows similar results: that the deformation matches low angle boundaries well. The red rectangular area in X0 shows a small misorientation of no more than 2°, and a parallel pattern, which can be observed in KAM as well, and suggests that the large grain cracked to release the energy so the further deformation stopped, because the stress did not concentrate in this region anymore. The white circle under the large grain is a characteristic of the migration and penetration of a grain boundary, which can increase the resistance for fatigue cracking [15], and it might be the reason why crack stopped near the boundaries below.

Different from initiation stage, fatigue crack tends to create large plastic deformation regions during propagation. According to IPF-Y0 in Figure 8, the fully deformed grain in Schmidt factor rotated to <111> during fatigue; the analysis of pole figure for the black rectangle area shows the same result: the grain rotates about 10° from boundary to inner area along the cyclic load direction. And it is common that the boundary area remains high in Schmidt factor while the center area is fully deformed, which implies the grain boundaries are barriers for deformation, due to the resistance of dislocation movement.

As is shown in Figure 8, the Schmidt factor varies a lot in different locations, even within a grain. However, the elastic modulus of aluminum in each orientation shows little difference, so a relatively uniform stress in each grain during fatigue was expected. This uneven deformation is caused by different mechanical properties of grains in rolled plate, affected by microstructures such as textures, second phase particles, grain boundaries, recrystallization, and many other factors. Therefore, uneven deformation resulting from those factors induces local stress concentration, which is the weakest spot for crack initiation and propagation. The uneven deformation idea is similar to that from the research by Li, L.L. et al. [15], wherein copper was studied by them; it has an anisotropic elastic modulus, which may cause uneven stress distribution and deformation. In addition, the research by Sangid, M.D. et al. promoted an increasing fatigue crack resistance as the Schmidt factor decreased [25], and that is a reasonable explanation for fatigue crack retardation during the propagation stage, perhaps a reason for why cracks initiate in the locations deforming unevenly as well. Overall, during deformation, the fatigue crack is obstructed by decreasing Schmidt factor in the plastic region ahead, while it tends to initiate and propagate in the uneven deformed location due to stress concentration; combined, this is a dynamic balance process.

### 3.6. EBSD Features of the Crack Tip in the Stir Zone

EBSD was acquired at the tip of fatigue crack in stir zone with an area of 60.9 × 41.8 µm by a step of 0.1 µm; the results are shown in Figure 10 and Figure 11. Comparing the two maps in Figure 10, there are some shadow areas. Some shadows are caused by a smooth transition zone between surfaces according to the SEI, because the diffraction pattern produced by a lower surface cannot be recognized. In the BC map, in the flat area there are some regions indicated by white arrows consisting of parallel lines with similar angles, which is a sign of strain filed. Also, there are some traces indicated with yellow arrows disappeared in the corresponding location in the SEI. And the orientation changed a lot between the two sides of a trace. Therefore, the traces were caused by lattice distortion and can be considered as initiating cracks.

In Figure 11, according to KAM, the plastic zone caused by crack is small, no more than 10 µm. In IPF-X0, Y0, and Z0, the orientation shifted little in deformed grains, and there was no obvious change in Schmidt factor. KAM shows a distribution concentrated mostly on grain boundaries, which implies a tendency of intergranular crack, considering the fact there is obvious difference between the orientation at both sides of most crack. Therefore, the intergranular crack propagates easier than a transgranular crack and consumes less energy.

The blue arrows in IPF-X0 indicate radial shape plastic regions induced as the crack attempts to propagate across grain boundaries. EBSD promotes a method to verify the simulation reports studied by Rice, J.R. et al. [29] and Kartal, M.E. et al. [30]. And the results about plastic zone near grain boundaries match with those researches to some extent. From the Schmidt factor a relatively long plastic region is shown in the yellow rectangular area, which can be observed in the KAM as well. A low angle boundary in this region is fading, as is shown in corresponding BC map. That low angle boundary decreased form 3.2° outside the region to 1.1° in the center. So, it can be assumed that during fatigue, the low angle boundaries can be eliminated as the plastic zone grows, and new ones parallel to load direction will be created. Besides, the blue arrows in Schmidt factor increase abnormally. That suggests a complicated stress condition, not demonstrated by cyclic force, is conducted in these regions.

## 4. Conclusions

This study produced proper specimens for EBSD observations of fatigue crack propagation stages by electrolytic polishing. The mechanism of fatigue propagation stage was studied. Based on the results and analysis, the following conclusions can be drawn:

(1) For AA 7N01, preparing an EBSD specimen with low voltage during electrolytic polishing can obtain high quality EBSD results in a large region. New possibilities for researching the mechanism of fatigue crack propagation are promoted with detailed features and a large region in this method.

(2) The resistance for fatigue crack propagation in FSW stir zone is lower than that in base metal. In stir zone, the crack has a tendency for intergranular growth with a smaller plastic zone. In base metal, the large fully deformed plastic zone may be the main reason for crack retardation and high resistance.

(3) Schmidt factor is an effective perspective to study material deformation from. Strain in the stir zone is concentrated with lattice distortion region due to strengthening effect of grain boundaries. The deformation region in base metal is large and continuous.

(4) Different factors that may affect the fatigue crack are observed simultaneously, including second phase particles, recrystallized grains, and grain boundaries. All of them can make contributions to a local strain incompatible.

The potential for further study of the fatigue mechanism using EBSD is presented in this research, especially for understanding and evaluating the influences of different factors simultaneously when focusing on a scale of hundreds or thousands of micrometers. The behavior and influencing factors of fatigue crack growth in friction stir welded joints still need more research.

## Figures and Tables

**Figure 1 materials-13-00330-f001:**
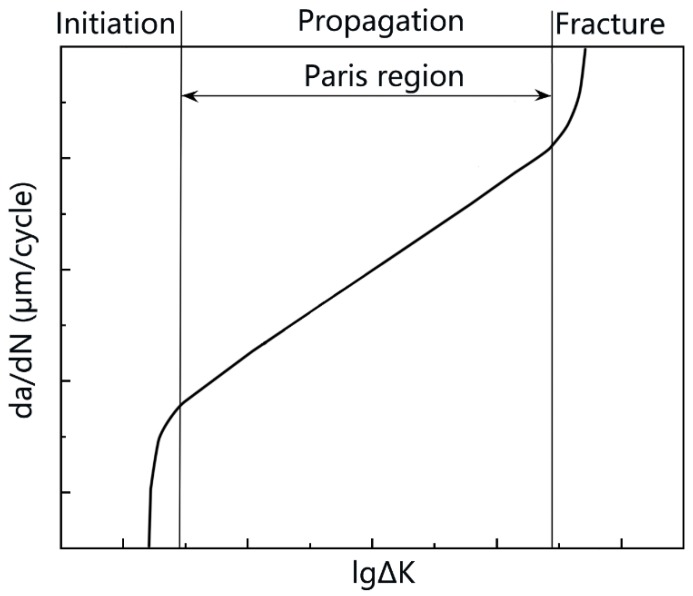
The different stages of fatigue crack growth. The Paris formula is suitable for describing the propagation stage.

**Figure 2 materials-13-00330-f002:**
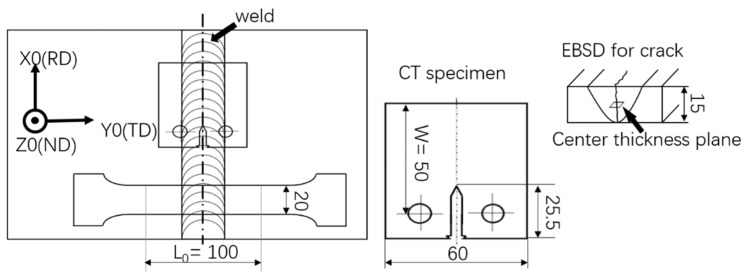
Schematic sample coordinate system; positions for CT (compact tension), tensile and EBSD specimens.

**Figure 3 materials-13-00330-f003:**
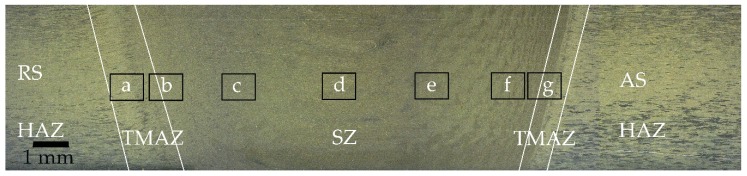
The macrostructure of the joint, showing different regions separated by white lines; the rectangles show EBSD acquisition spots.

**Figure 4 materials-13-00330-f004:**
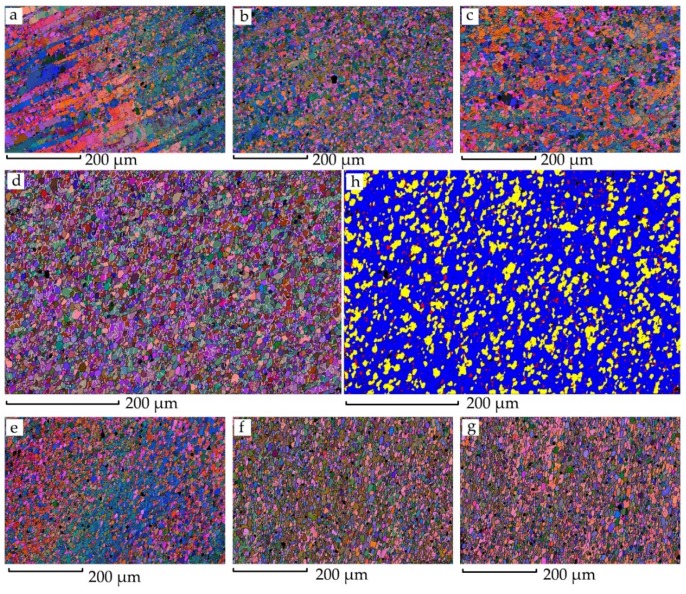
EBSD results of the joint. Image (**a**–**g**) correspond to the rectangle regions in Figure 3. Image (**h**) is the recrystallized fraction result of the region (**d**).

**Figure 5 materials-13-00330-f005:**
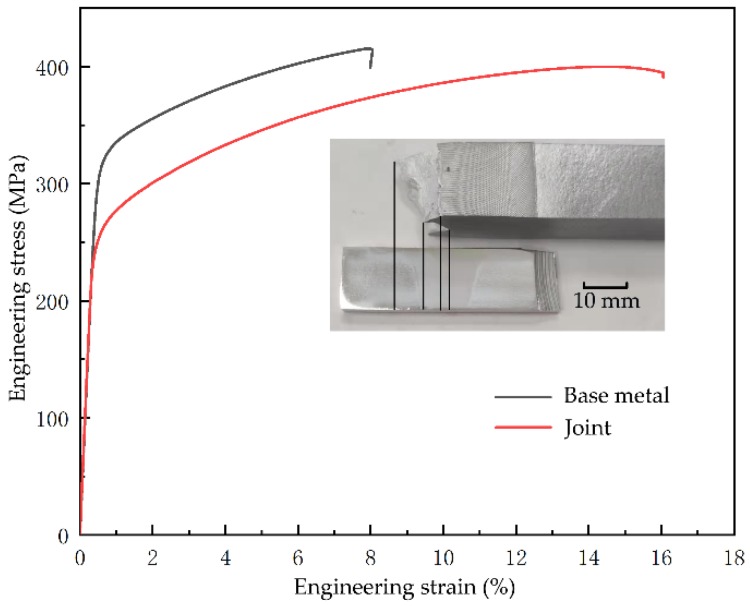
The curve of engineering stress–engineering strain, and the position of fracture.

**Figure 6 materials-13-00330-f006:**
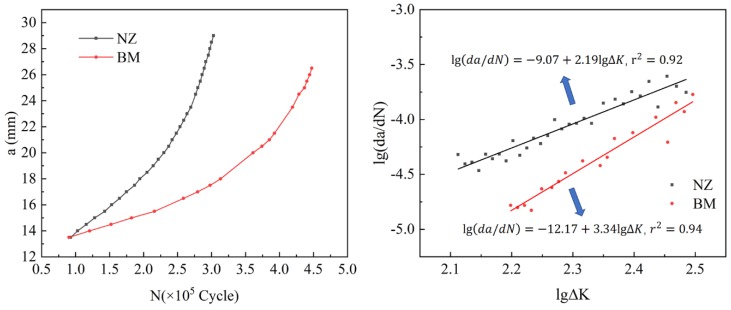
Results for FCP (fatigue crack propagation) test, a–N curve (**left**), and lg(da/dN)-lg∆K curve (**right**).

**Figure 7 materials-13-00330-f007:**
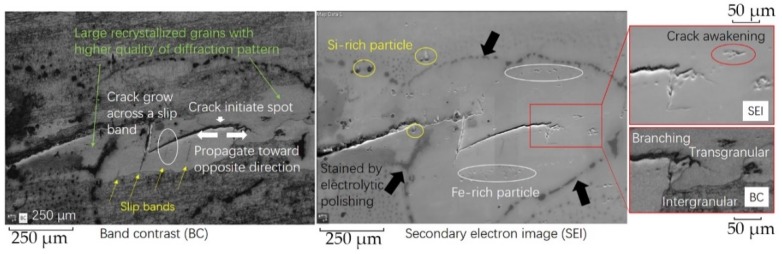
BC (band contrast) map and SEI (secondary electron image) at the crack tip on the base metal.

**Figure 8 materials-13-00330-f008:**
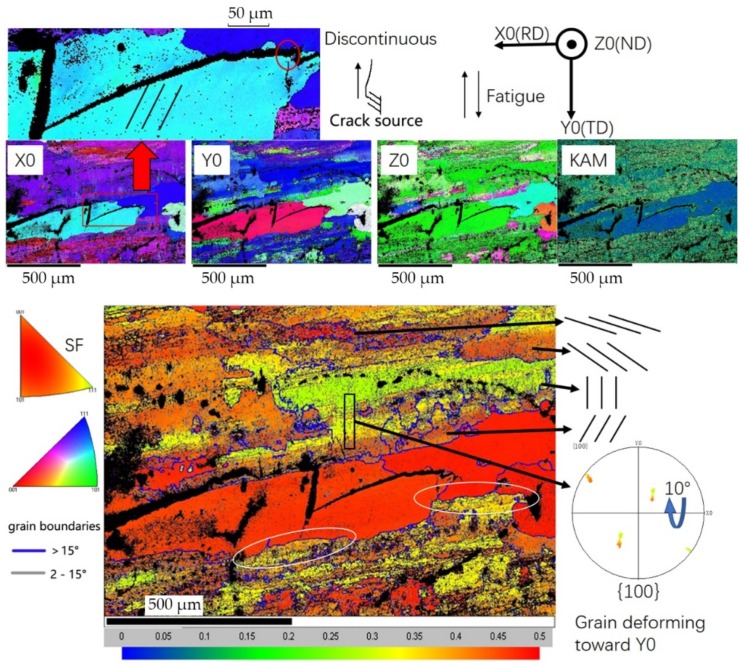
Maps of IPF-X0, IPF-Y0, IPF-Z0, KAM (local misorientation), and Schmidt factor near the crack in the base metal.

**Figure 9 materials-13-00330-f009:**
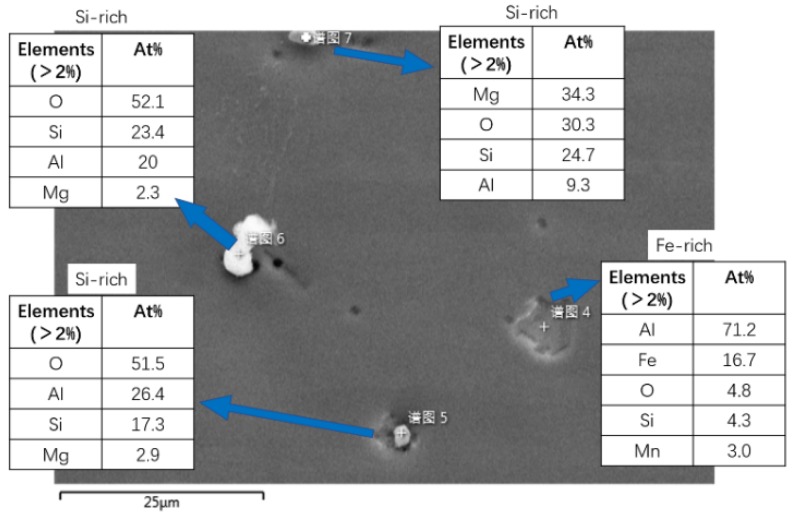
EDS (energy dispersive spectrometer) characteristics for different second phase particles in base metal.

**Figure 10 materials-13-00330-f010:**
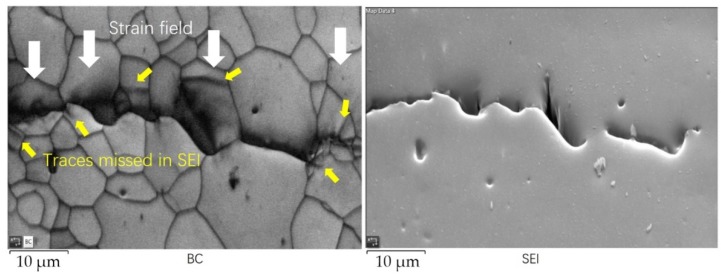
The BC map and SEI at the crack tip on the base metal.

**Figure 11 materials-13-00330-f011:**
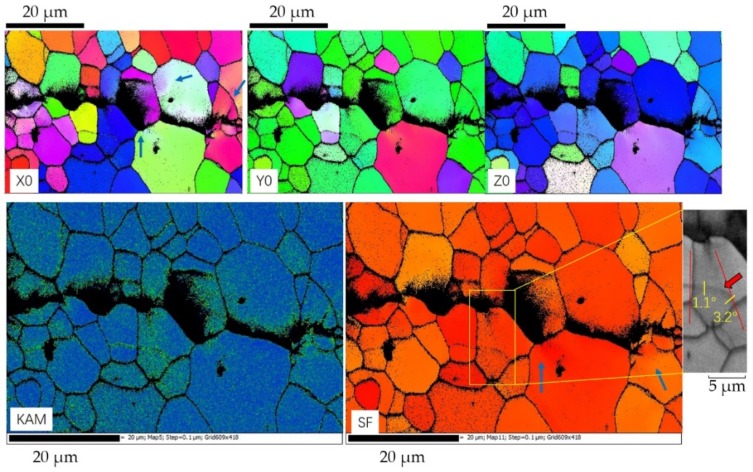
Maps of IPF-X0, IPF-Y0, IPF-Z0, KAM (local misorientation), and Schmidt factor near the crack in stir zone.

**Table 1 materials-13-00330-t001:** Chemical composition of AA 7N01.

Elements	Si	Fe	Mn	Cu	Cr	Zr	Mg	Zn	Al
**wt%**	0.230	0.189	0.343	0.038	0.090	0.107	1.23	4.02	Bal.

**Table 2 materials-13-00330-t002:** Tensile properties of base metal and the joint.

Specimens	Ultimate Tensile Strength (MPa)	Yield Strength (σ_0.2_) (MPa)	Elongation (%)
**Base metal**	415	320	7.4
**Joint**	400	256	15.5

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
