# Peer review of "A Study on Fatigue Crack Propagation for Friction Stir Welded Plate of 7N01 Al-Zn-Mg Alloy by EBSD"

_materials, 2020, doi:10.3390/ma13020330_

Round 1

Reviewer 1 Report

Dear authors!

First of all, I want to note my own positive impression of the article. It contains relevant and valuable results, especially in the field of deformation zone sizes evaluation in the fatigue crack tip.

There are a few suggestions that may improve the article.

The article includes the study of plastic deformations and fracture of the 7N01 alloy, so the information about its mechanical characteristics, especially on the strain hardening ability and ductility, should be performed. Stress-strain curves and/or a table with mechanical properties would be very helpful. The sentence “However, the assessment ...” (lines 39-41) obviously lacks a verb. Line 72. I suppose that W-parameter should be marked in Figure 1 (specimen scheme). Line 189. Ref to Michael D research is performed, but ref [21] is the research of Mr. Sangid M.D. et al. Please check if everything is OK with this sentence. Conclusion #4 (lines 246-248) is too blurred and contains well-known information. More detailed interpretation is required.

Author Response

Thank you for the good suggestion.

Point 1: The article includes the study of plastic deformations and fracture of the 7N01 alloy, so the information about its mechanical characteristics, especially on the strain hardening ability and ductility, should be performed. Stress-strain curves and/or a table with mechanical properties would be very helpful.

Response 1: The chemical composition of 7N01 is listed. The stress-strain curve of the base metal and the joint is presented now.

Point 2: The sentence “However, the assessment ...” (lines 39-41) obviously lacks a verb. Line 72. Line 189. Ref to Michael D research is performed, but ref [21] is the research of Mr. Sangid M.D. et al. Please check if everything is OK with this sentence.

Response 2: We have deleted the sentence “However, the assessment ...”. We added two sentences “However, the mechanism of fatigue is not fully understood. The fatigue of materials after friction stir welding still needs more research because of the varied microstructure of the joint.”. In order to make the introduction clearer and more oriented.

Point 3: I suppose that W-parameter should be marked in Figure 1 (specimen scheme).

Response 3: W-parameter is marked in specimen scheme in revised manuscript.

Point 4: Line 189. Ref to Michael D research is performed, but ref [21] is the research of Mr. Sangid M.D. et al.

Response 4: The ref. to “Michael D” has been revised to Sangid M.D. et al.

Point 5: Conclusion #4 (lines 246-248) is too blurred and contains well-known information. More detailed interpretation is required.

Response 5: Conclusion #4 has been changed to “Different factors may affect the fatigue crack are observed simultaneously, including second phase particles, recrystallized grains and grain boundaries. All of them can make contribution to a local strain incompatible.”. We think that is a more specific interpretation of our intentions.

Reviewer 2 Report

The paper (i.e. the study itself) is clear and well focused, the conclusion effectively wraps up and goes beyond restating the thesis. Writing is smooth, skillful and coherent. The sentences are strong and expressive with varied structure. The diction is consistent and words, figures ad tables are well chosen. 

From the point of view of my scientific interests - the reviewed article is very interesting and at a good scientific level. I have one suggestion for the authors: each method has its limitations. In this case, are we dealing with the limitations of the EBSD method in the aspect of fatigue crack propagation testing? If so, this would be very important information.

Author Response

Thank you for the good suggestion.

Point : I have one suggestion for the authors: each method has its limitations. In this case, are we dealing with the limitations of the EBSD method in the aspect of fatigue crack propagation testing? If so, this would be very important information.

Response : We think the main limitation of observing fatigue with EBSD lies in two points:

Preparing EBSD specimens is a destructive process no matter what method is used. We made specimens with careful grinding and polishing to minimize the damage. And we think the results still keep some important information according to the reasonable details we observed. Fatigue behavior is different throughout the thickness direction. We observed the plane of the center thickness. Because this plane is the most constrained one and can be considered as plain strain state, making the test repeatable. During specimen preparation, each grinding and polishing will show results with little difference due to the tens of micrometers material removed. We got several results and presented the most representative ones.

Reviewer 3 Report

Its an interesting study about Fatigue Crack Propagation for Friction Stir Welding.

However the reviewer is not a native English speaking person, the language  needs serious improvement, there are sentences very hard to understand.

Generally:

Some major parameters are missing, e.g. base material composition, mechanical properties. and alla the FSW parameters figures need some improvement, scale bars are hard to read, the font sizes must be unified. please give some hint where the results of this research can be used, and how can a welding engineer (which I am) use the results of this paper!

I made my specific remarks, questions, comments in the manuscript_with_reviewers_comments

With proper corrections I think the manuscript maybe satisfy the publication criteria in Materials.

Author Response

Thank you for the good suggestion.

Point 1: Some major parameters are missing, e.g. base material composition, mechanical properties. and alla the FSW parameters figures need some improvement, scale bars are hard to read, the font sizes must be unified.

Response 1:

In the revised manuscript, the chemical composition of 7N01 is listed. The FSW parameters is added. The stress-strain curve of the base metal and the joint is also presented. We added the fracture position as well.

We have adjusted all pictures except the one showing fatigue results. Every picture has a proper scale bar now. The font size is unified as well.

We have corrected some typos and wrong sentences. Including the sentences marked in the reviewed PDF and some other sentences may hard to understand.

We have changed “most previous reports” to “many previous reports”, and added several references here (line 156).

Point 2: please give some hint where the results of this research can be used, and how can a welding engineer (which I am) use the results of this paper!

Response 2: This study is just a primal understanding of fatigue propagation in FSW joint. We believe it will contribute to controlling and improving welding quality especially for analyzing failure mechanisms in further research.

Reviewer 4 Report

GENERAL COMMENT

The subject matter of the article has some significance and may arouse interest, but the preparation of the article has not been carried out at the appropriate level. The article has a correct structure, but the introduction is short and shows a poor orientation of the authors in the topic being addressed. The material and methodology do not contain the most important information about the material to be tested, how the joint is made and the dimensions of the notch of the test specimens. In general, the interpretation of results and illustrations contains significant imperfections, which need to be corrected and supplemented. Before publishing the article, the descriptions should be completed, drawings and photos should be corrected, so I propose to reject the article in its current form.

The most important shortcomings

- lines 26-56 There is no reference to specific literature items, and several items have been collected collectively, eg [5-10] or [11-17]

- lines 57 -87 Lack of material description, conditions of FSW connections, notch geometrical parameters. Please explain, why were native material and nugget zone tested under different conditions?

- line 97 Figure 2. Poor quality photo

- line 124 Figure 4. It is not allowed to compare the results of tests carried out under different conditions (lines 76-77). In addition, the drawing on the right is incomprehensible. Fig.4 (right) includes only a fragment of the crack propagation graph.

- line 156 Figure 6. The drawing is illegible.

lines 158 – 248 Autors in the article uses the Schmidt factor without prior explanation and the values are not strictly stated. The analyses are based on two photos, without reference to at what stage of crack propagation they were made and at what stress concentration factor.

Author Response

Thank you for the good suggestion.

Point 1: lines 26-56 There is no reference to specific literature items, and several items have been collected collectively, eg [5-10] or [11-17]

Response 1:

The introduction has been adjusted with adding more description to references, to make the introduction more oriented.

More description to references is added into the introduction. The reason for the collected references is that we sorted those studies by the method they used. So, many studies employing similar method was put together.

Point 2: lines 57 -87 Lack of material description, conditions of FSW connections, notch geometrical parameters. Please explain, why were native material and nugget zone tested under different conditions?

Response 2:

The chemical composition of 7N01, FSW parameters and the notch length are added. The stress-strain curve of the base metal and the joint is presented now.

The reason for different load force is explained in the revised manuscript (line 86 to 96). Considering the lower yield strength of the joint, we set a lower force for nugget zone to save time. And according to experience, both the base metal and nugget zone are in the propagation stage, where the Paris formula is suitable. Then, all the parameters and results are processed into log(da/dN)-logΔK. It is the constant C and m determined by materials that influence the Paris region.

In addition, the ASTM E647 writes: Expressing da/dN as a function of ΔK provides results that are independent of planar geometry, thus enabling exchange and comparison of data obtained from a variety of specimen configurations and loading conditions.

Point 3: line 97 Figure 2. Poor quality photo

Response 3: This photo has been changed to a higher quality one (Figure 3 in revised version).

Point 4: line 124 Figure 4. It is not allowed to compare the results of tests carried out under different conditions (lines 76-77). In addition, the drawing on the right is incomprehensible. Fig.4 (right) includes only a fragment of the crack propagation graph.

Response 4:

The reason for the different load force has been explained above. So we think comparing data in the same stage (propagation) with similar ΔK is reasonable.

Limited by the specimen length and the requirement:, it is very hard to obtain fatigue data with a larger ΔK range. Usually, the testing for whole curve of the fatigue, from initial stage to fracture stage, requires lots of different fatigue experiments.

Point 5: line 156 Figure 6. The drawing is illegible.

Response 5: The picture is changed to a clearer one (Figure 8 in revised version) to explain our point.

Point 6: lines 158 – 248 Autors in the article uses the Schmidt factor without prior explanation and the values are not strictly stated. The analyses are based on two photos, without reference to at what stage of crack propagation they were made and at what stress concentration factor.

Response 6:

Figure 9 is added to explain Schmidt factor. Due to the lack of knowledge to strictly state the values of Schmidt factor, we can only refer to the distribution map made by EBSD analysis software Channel 5. And the results seem reasonable.

The ΔK at the crack shown in EBSD is added to line 100 to 102.

Round 2

Reviewer 3 Report

The manuscript was improved significantly from the initial state, most corrections and answers to my questions were adequate, some issues remain open:

Fig 1 has very large fontsize

Fig 3 EBSD images have too small magnification, please enlarge

Fig 5 has different fomnt type

Table 2 values should have +- scatter, how many aspecimens were tested?

fig 9 schematic drawing for Shmid factor is trivial (recommend to cut it from manuscript)

A comparison of the results with literature data would be great, at least for the main mechanical prop. UTS, Fracture elongation, hardness profile...

With proper corrections I think the manuscript can satisfy the publication criteria in Metals.

Author Response

Thank you for good suggestion.

Point 1: Fig 1 has very large fontsize

Response 1: The font size has been changed smaller.

Point 2: Fig 3 EBSD images have too small magnification, please enlarge

Response 2: The images have been enlarged. And the description has been changed to a more proper one.

Point 3: Fig 5 has different fomnt type

Response 3: The font type has been changed.

Point 4: Table 2 values should have +- scatter, how many aspecimens were tested?

Response 4: We have added the scatters. We tested two specimens for base metal and joint respectively and the values are the average of them.

Point 5: fig 9 schematic drawing for Shmid factor is trivial (recommend to cut it from manuscript)

Response 5: This picture has been removed.

Point 6: A comparison of the results with literature data would be great, at least for the main mechanical prop. UTS, Fracture elongation, hardness profile...

Response 6: We have added a reference that has a similar condition with our study and made a comparison of the tensile results.

Reviewer 4 Report

Dear Authors
Thank you for making numerous corrections that significantly improve the quality and transparency of work.

Author Response

Dear reviewer

Thank you very much for the reviewing.